# Spatial impacts of the intercity flow of long stay international visitors based on mobile phone data in Yangtze River Delta, China

Yao Wang[☉], Meilin Zhu[☉], Xiaodong Meng[ID]*[☉]

School of Architecture and Urban Planning, Suzhou University of Science and Technology, Suzhou, Jiangsu Province, China

☉ These authors contributed equally to this work.

* 245540789@qq.com

**Data Availability Statement:** All relevant statistical data are within the paper and its Supporting Information files. The authors do not own the mobile phone data. Data are available from China Unicom's operators. Researchers can contact the

## Abstract

The intercity daily flow of long stay international visitors (LSIV) reflects the economic globalisation and regional integration of a region. We made the first attempt to use mobile phone data to identify international visitors who stayed longer than one month in the Yangtze River Delta (YRD) region in 2019, analysed the spatial patterns of LSIV, and revealed the impact factors associated with the daily intercity flow of LSIV. We discussed spatial dependence using multiscale geographically weighted regression (MGWR), and performed cluster analysis to understand the combination effects. The results show that enterprises have the largest effect, AQI and administrative hierarchy have relatively low effects, and income, imports and exports do not have any significant effect. Overall, the economically developed eastern regions of the YRD region are more attractive for daily travel by LSIV, with the Shanghai metropolitan area being the most attractive. Our findings provide new insights into the relationship between the intercity daily flow of LSIV and the urban economy and society in the delta region to help suggest planning recommendations to enhance the globalisation development strategy and provide a better environment for international visitors in the delta region.

## 1. Introduction

China's opening to the outside world has attracted global capital and multinational companies to its coastal regions, which has significantly contributed to economic growth and urbanisation since 1978 [1]. Along with multinational corporations, there is a large flow of international visitors, including business people, managers, technicians, and tourists. The number of international visitors to China as a destination has grown at an average annual rate of 11.96% for more than four decades, reaching 65,725,200 in 2019 (Statistical Yearbook of Chinese Cultural Relics and Tourism) before the COVID-19 outbreak. The flow of international visitors builds trusting relationships through face-to-face contact, bringing business opportunities and advanced technology, and spreading knowledge that can drive economic growth in destination countries, which cannot be replaced by new communication technologies [2, 3]. International visitors often stay in the arrival country for a long time for business work, study, tourism and

company for data applications (https://gec.10010.com/ln/product/1001342).

**Funding:** This work is supported by the National Natural Science Foundation of China (NSFC, https://www.nsfc.gov.cn/) grant 52008281. The funders had no role in study design, data collection and analysis, decision to publish, or preparation of the manuscript.

**Competing interests:** The authors have declared that no competing interests exist.

other purposes, and their choice of arrival city is more purposeful and diverse. It is worth noting that compared to short stay international visitors, the intercity daily flow of long stay international visitors (LSIV) within the destination country has a more pronounced impact on business opportunities, advanced technology, and knowledge dissemination. The impact on urban economic and social development has been sustained. Therefore understanding the spatial relationship between intercity flow of LSIV and urban socio-economic development within a country region, that is, what social attributes of cities attract the flow and stay of LSIV, is of interest for the study of intra-regional economic development differentiation.

Studies on human mobility in a country or region focus on two patterns. One is population migration, which is the permanent or temporary movement of people over long distances and long periods of time to new locations, including urban-rural migration, intercity migration, and international migration [4]. Herberle's "push-pull" theory explains migration as the result of a combination of "push" forces from the place of departure and "pull" forces from the place of entry, such as residents from less-developed cities moving to developed metropolitan areas for better employment, education, health services, or other benefits [5]. The other type of movement is intercity daily flow, in which residents travel daily for work, business, consumption, tourism, or entertainment intercity [6, 7]. In view of the development of high-speed transportation technology and mobile communication facilities, the flow of people, capital, information, and other factors accelerates, and space becomes dynamic and networked, Castells proposed the "Space of Flow" theory [8]. The intercity daily flow of people is actually a short-term and high-frequency travel behaviour of residents from the origin to the destination, and the resulting intercity people flow network is in line with the characteristics of the Space of Flow theory [9]. Since these two types of mobility patterns have different theoretical foundations, actors, mobility characteristics, and socio-economic bases, the two different fields of literature are not obviously interrelated.

In this paper, we focus on LSIV as foreign visitors who stay in the destination country for more than one month, and the travel of this group of people is a short-term transnational migration behavior at the global scale. However, at the regional scale it is essentially more similar to the intercity daily activities of LSIV. Although some literature has used the "push-pull" model constructed by Population Migration theory to explain intercity daily mobility [10], this paper argues that the migration of international visitors at the global scale is a distinctly different scale and issue from intercity mobility within regions. It is clear that the spatial theory of mobility is better suited to explain this pattern of mobility and is more consistent with the mechanisms by which "urban attribute" drive and influence "the network of flow".

From the current research, many scholars at home and abroad have studied mobility from the perspectives of both external representations and internal causes. The external representation perspective focused on the concepts and spatial characteristics of mobility, such as the spatial structural characteristics of mobility, the behavioral mechanisms of mobility, and various characteristics and attributes [11, 12]. From the perspective of internal causes, they have analysed the influencing mechanisms of mobility, such as the macro build-up factors of cities or the micro individual decisions factors [10, 13]. The research content mainly focuses on the urban network pattern of daily travel of the country's residents, but the role played by urban attributes in the mobility process has been less explored. And apart from the Space of Flow theory to explain the intercity daily flow of international visitors, the literature on the intercity daily flow of international visitors has paid less attention to explaining the spatial heterogeneity of mobility. That is, various geographical differences are still considered as complex motives for the combination of intercity daily flow, and explaining spatial differences helps to understand the relationship between intercity flow and related socio-economic development [14, 15]. To the best of our knowledge, such studies have not been published. Therefore, this study attempts to

establish differences in spatial patterns that reveal daily intercity movements of LSIV, explain socio-economic factors associated with mobility patterns, and fill relevant research gaps.

The flow of international visitors used in previous studies was based on the Statistical Yearbook and survey data. Although Statistical Yearbook data reflect changes in the number of international visitors in a given region, they do not reflect the spatial distribution and flow characteristics of international visitors. Survey data based on small samples may not accurately reflect the dynamic changes and may not provide a comprehensive understanding of the urban agglomeration scale [16, 17]. China's current official tourism statistical yearbook only records the total number and source countries of international visitors entering China for customs statistics and does not record in detail the intercity travel of international visitors within China. There has been little effort to quantify the destinations of international visitor flows within China through large-scale mobility observations.

Along with the emergence of Big Data, studies have commonly used global positioning systems (GPS), social media, mobile phone signalling, and other mobile location. Big Data have been widely used to extract human activity data, often including geographic information, which can be used to identify patterns of large-scale intercity travel [18, 19]. From a data acquisition perspective, GPS data collection is expensive and time-consuming and is primarily used to understand visitor mobility around destinations (e.g. cities) or attractions (e.g. parks) [20]. For example, social media data (e.g. geotagged photos) allow for visitor travel analysis on a broader scale and can reveal rich contextual information about visitors. Nevertheless, such data are temporally and spatially sparse and irregular [21]. Mobile phone signalling data features a large sample size, high temporal continuity, and high accuracy, which can be used to identify the large-scale spatiotemporal movement of users [17, 22]. By capturing the mobile phone signalling signals of foreign visitors through mobile service companies, we obtain location space information similar to that of domestic users, making it ideal for identifying the intercity mobility of LSIV. Although the data does not contain information on individual social attributes, such as age, gender, and education, it is able to capture the location footprint of a large population at a large scale [23].

By using international mobile phone roaming data, we can effectively identify the travel patterns of LSIV. Thus, firstly, using international mobile phone roaming data we can effectively identify the spatial patterns of LSIV's intercity daily flow, then we reveal the socio-economic factors associated with the spatial patterns using a multiscale geographically weighted regression (MGWR) model, and finally, perform a cluster analysis. The contributions of this study are fourfold: First, we develop an analytical framework for an "attribute-network" model based on the Space of Flow theory to explain how urban attributes affect the flow networks of LSIV. Second, we propose a new method for large-scale identification of LSIV and construct an evaluation model of various urban economic and social indicators on the intercity flow of LSIV. Third, we find through spatial heterogeneity analysis that there are large spatial differences in the influencing ability of each indicator and the driving mechanism of intercity flow differs from traditional perceptions. Fourth, we found through cluster analysis that the stronger differences in the attraction of long-term stay international visitors in different regions of the Yangtze River Delta are characterized by significant spatial imbalance, which may be related to the spatial differences in the economic and social development of cities.

## 2. Literature review

### 2.1 Population migration theory and space of flow theory

Population migration is the movement of people between two regions, generally involving long-term changes in place of residence, and can be divided into domestic and international

migration depending on the destination. Ravenstein (1885) first introduced the "The Laws of Migration", which suggests that migration is driven by economic incentives, with people being pushed by unfavourable conditions at their place of origin and pulled by attractive conditions at their destination, thus forming the "push-pull" theory [24]. Lewis (1954) proposed the "Dual Sector model", which suggests that there is a directional migration of surplus rural labor to cities in developing countries [25]. Zelinsky (1971) was the first to study the international migration of people, combining the size and flow of people leaving their home countries with the socio-economic modernization process, and found a clear pattern of international migration [26]. Population Migration theory, which focuses on the patterns of change of people at temporal and spatial scales, has been emphasizing the direction of migration and decision-making mechanisms.

Space of Place theory assumes that economic activity and the movement of people are strictly limited by geographical space and distance, emphasising locality. With the rise of contemporary networked societies where connections cross geographical boundaries, Castells' Space of Flow theory differs from the traditional Space of Place in believing that the revolution in communication and information technology has made the space of flow, which is composed of information, capital, and technology, the dominant spatial form [8]. Real "flows" represented by the Relational Data are used as functional links between cities. This brings about a shift in the underlying theory, external form, and internal mechanism of regional spatial structures, which Meijers (2007) sees as a paradigm shift in regional research [27]. Taylor (2010) integrates the space of place and the space of flow and further proposes the Central Flow theory, which argues that Central Place theory should not be completely rejected and should focus on the external connections of cities [28]. The Space of Flow theory is introduced into the study of urban or regional space, shifting the study from the internal characteristics of cities to the external relations of cities, and the focus from the morphology, core periphery, and hierarchy of cities to the structure, function, and connection relations of urban networks.

The similarities between Population Migration theory and Space of Flow theory both focus on changes in the spatial location of residents and the mechanisms that influence them. Migration theory emphasises the 'push and pull' of the place of origin and the place of migration, and focuses more on the impact of changes in residence on urbanisation, urban patterns and social equality. The Space of Flow theory is more concerned with the impact of the flow of various "flow" elements, such as human, logistics and information flow, on the spatial structure of regions and cities. This paper focuses on long stay international visitors as a type of short-term international migration that transcends geographical constraints and is characterised by international migration behaviour, as well as by the pull of the "flow element" within the region of the country of destination, and is characterised by both migration theory and the Space of Flow theory.

## 2.2 Residential migration and human flow

Regional imbalances in development are an important incentive for migration, and migration so that migration has a distinctly economic character. Disadvantages such as scarce employment, education and health facilities in less developed regions drive the population to move to developed regions for long periods and long distances. Todaro (1969) argues that population migration is usually accompanied by the transfer of young and middle-aged labour, which provides sufficient labour for the incoming area while promoting the concentration of talent and industry and optimising the allocation of production factors, and that the local urbanisation of population rapidly accelerates endogenous local economic growth and the expansion of urban patterns [29]. For the individual migrant, moving to a developed region not only increases

wages, but also the accumulation of professional skills, the level of education of children and the access to life services [30]. At the same time, the financial and intellectual capacity acquired by migrants also contributes in turn to the development of the home countries that receive remittances, resulting in stronger economic, social and information links between the incoming and outgoing areas [31].

Urban networks formed by inter-regional flows of people, capital, information, and technology are characterised by diversity. Among the flourishing theoretical and empirical studies, different types of real "flows" reflecting the characteristics of urban networks have become the mainstream of research [9, 32]. Although the different agents that generate connections correspond to different spatial structures and driving mechanisms of intercity connections, this study focuses on the real flows of people. Human mobility is undoubtedly considered to be one of the most important lenses through which intercity mobility is observed, as it drives more frequent spatial clustering and diffusion of various factors, laying the foundation for capital flows and knowledge exchanges [6]. The Central Place theory proposed by Christaller (1930) incorporates the Economic Man assumption of classical economics and is constrained by the technological conditions of time to consider only the accessibility of residents and the range of urban goods and services, which is essentially the movement of people represented by the travel of residents [33, 34]. Hall (2006) argues that intercity travel is a spatial continuation of business and social relations, reflecting the functional links between cities in business and trade [35].

According to the "push-pull" theory, population migration occurs mainly as a result of regional socio-economic differences. Rational and orderly population migration is an important factor in optimising the production structure and promoting socio-economic development, and has an obvious contribution to the development of urbanisation and social and cultural integration in the places of migration to and from. Unlike migration due to economic differentials, intercity human mobility is a high frequency of residential travel between neighbouring spatial units and includes diverse purposes such as commuting, business, leisure, consumption and tourism. Regional development differences lead to differences in the external functions of cities, and the intercity daily mobility of residents to meet their own personal needs for work, life and hobbies reflects the derived needs of residents to participate in activities at different points in time and space[36]. The aforementioned needs of residents are far from being such that they have to decide to move and can be met by intercity daily flow, so there are significant differences in the nature and motivation of intercity daily flow and migration. Therefore, this paper focuses on international visitors who stay in China for more than one month. The movement of these people, who arrive and stay in China permanently from various countries, is migration, but their travel within the Yangtze River Delta region is not an act of relocation but an intercity movement for various needs, and their daily movement enhances the intercity links of various elements.

## 2.3 Impact factors of residential migration and human flow

Whether within countries or internationally, the 'push' from the place of departure and the 'pull' from the place of entry have always been seen as the main causes of migration. Push factors include social, economic and political forces such as low wages, scarce jobs and high house prices; pull factors include better job opportunities, a prestigious business environment or a better quality of life [37]. It is clear that inter-regional migration is often related to the economic environment [38], with good economic conditions in developed regions providing more jobs and higher pay conditions to attract people, while better educational resources in larger cities attract young migrants [10, 39]. Migration behaviour may also be influenced by

individual intentions. Self-attributes such as gender, age, marital status and educational attainment may also have an impact on the decision to migrate [40]. However, it has been found that individual socio-cultural factors are much less influential on migration decisions than economic factors such as income and employment in terms of overall motivation to migrate [41].

The determinants of intercity daily flow can be divided into two major categories according to the research scale: one is the macro environment, such as regional economic and social, built environment, and government resources, and the other is the micro individual decisions of individual social attributes and behavioural styles. Macroenvironment-related factors play an important role in promoting intercity daily flow, similar to the influence of the push-pull model, including economic development, residents' income, foreign investment utilisation, urban construction, ecological environment, infrastructure, government resources, and administrative power [13, 42–44]. Micro individual decision making involves the subjective judgment of individuals in the process of travel occurrence, which is not only affected by individual social attributes, such as gender, age, occupation, income, and emotion, but also involves individual behavioural preferences of transportation modes and transportation costs [15, 45, 46]. Generally speaking, the study of factors influencing intercity flow is related to the scale of research, and large-scale studies, such as regional and urban areas, start from the macro environment. Therefore, this paper builds on this idea.

Population migration is motivated by the pursuit of higher economic benefits, with more attention paid to factors such as employment, housing and social security, as well as the overall economic climate, which are closely linked to the fundamental livelihood security of the population. The daily mobility of the population focuses on factors such as interests, modes of transport, weather and the built-up environment. The factors influencing mobility are more integrated and comprehensive, and the differences in individual choices are more diverse. Thus, in the context of globalisation, the intra-country or intra-regional mobility of long stay international visitors is closer to intercity daily flow and the influences on space of flow better explain the intercity mobility of long stay international visitors.

## 3. Data and methods

### 3.1 Conceptual model

Although the Space of Flow theory emphasises the flow of people, capital, and information between cities to form urban networks, Taylor (2010) finds that the flow of factors within a region is not separated from the space of place, but rather works together with the interaction of mobility space. Existing research takes enterprises as the object of study and explores the interaction between enterprise agglomeration behaviour and enterprise association network in the process of enterprise organisation, forming an internal logic of unifying regional attribute and urban network, which is summarised as the "urban attribute derives from the "urban network", and the "urban network" is rooted in the "urban attribute" [47, 48]. In other words, the "urban attribute" of population, industry, market, and other factor concentration sites and the "urban network", formed by the flow of people, capital, and information, are in close interaction, and the differences in the attributes of different sites play a key role in the flow of factors. Therefore, to investigate the factors influencing the spatial patterns related to the intercity daily flow of LSIV, this study constructs a conceptual framework of "attribute-network" (Fig 1).

Under the theoretical framework of "attribute-network", we have built a dataset focusing on three city attributes, namely, economy, urban built environment, and government supply resources (Table 1). This network describes the economic, political, transportation, environmental, and technological characteristics of 190 cities in the YRD region, and comprises a fairly complete dataset.

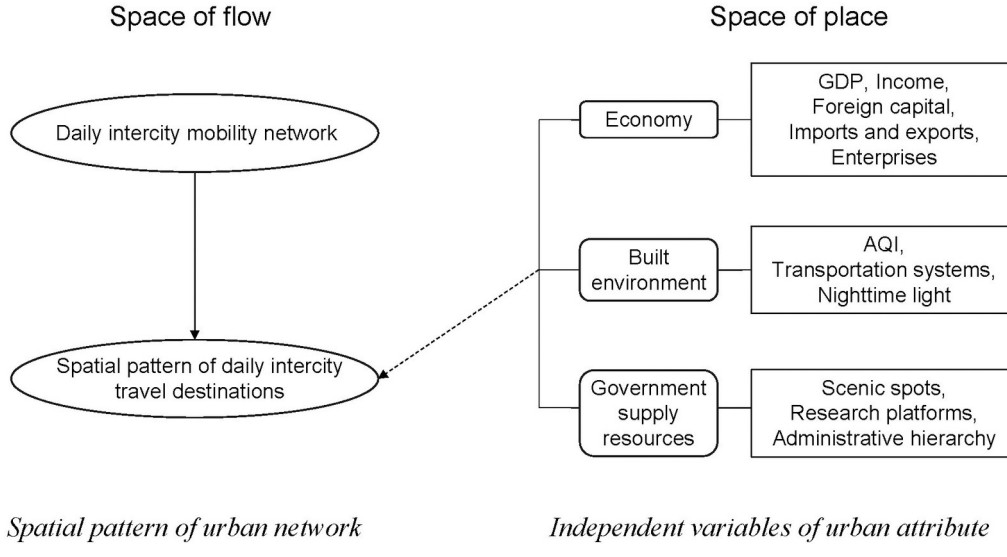

Space of flow

Space of place

*Spatial pattern of urban network*

*Independent variables of urban attribute*

**Fig 1. Conceptual framework.**

Economic factors are important indicators of population migration flow, which include statistics such as gross domestic product of the region, urban and rural per capita disposable incomes, utilised foreign capital, total volume of imports and exports, and the number of World Top 500 Headquarters, Branches and Mutual Investment Enterprises. GDP is an important indicator of a region's economic situation and development level, and is a frequently used economic variable. Income represents the ability of residents to dispose of leisure and recreational behaviours, such as shopping, recreation, and tourism, which have a significant impact on residents' travel [49], Cities with higher urban and rural per capita disposable incomes may be more attractive to international visitors for intercity mobility. The participation of cities in global production networks is reflected in the actual utilisation of foreign capital and the city's total volume of imports and exports. Cities with high foreign trade are attractive to LSIV [15]. Work and business have always been the driving force behind intercity travel for local residents [50], and along with economic globalization, international business

**Table 1. Data summary of the selected variables.**

| Category | Variable | Description |
|---|---|---|
| economy | GDP | Gross Domestic Product ($10^8$ yuan) |
| | income | urban and rural per capita disposable incomes (yuan) |
| | foreign capital | actually utilized foreign capital ($10^8$ dollar) |
| | imports and exports | total volume of import and export ($10^8$ dollar) |
| | enterprises | the number of World Top 500 Headquarters, Branches and Mutual Investment Enterprises |
| urban built environment | AQI | air quality index |
| | transportation systems | whether the city has high-speed railway, airports and whether the average road network area is higher than regional average |
| | nighttime light | total brightness value of nighttime light (DN) |
| government supply resources | scenic spots | the number of national 5A-level tourist attractions |
| | research platforms | the number of country-level scientific research platforms |
| | administrative hierarchy | rank by municipality directly under the Central Government, sub provincial level/city specifically designated in the plan, provincial capital, general prefecture level city/county directly under the provincial government, and county directly under the municipal government |

travel is driven by the global presence of multinational enterprises [2]. The efficient operation of multinational enterprises requires the stable services of foreign management and technical staff, and attracts international visitors for long-term stays and intercity mobility. Thus, the number of multinational enterprises has become a key factor in attracting long-term international visitors.

Urban built environment factors are directly related to the level of urban construction and development, reflecting the city's development intensity, ecological environment, and infrastructure, which may have an impact on the intercity daily flow of the LSIV. Air quality can affect the health of residents, particularly since the deterioration of air quality due to industrialisation and urbanisation in China is a growing health concern [44]. Transportation connectivity plays a fundamental supporting role in population mobility, with high-speed rail and airlines serving long-distance travel and being the main modes of intercity daily flow for the international visitors [51]. Nighttime lighting data are a valid indicator of urban environmental development intensity, and studies have shown a positive association between urban environmental development intensity and long-distance travel behaviour [52].

Government supply resource factors refer to the government's direct or indirect support for the development of various types of publicly owned undertakings through funding. The abundance of tourism resources in a city directly affects the movement of people for tourism purposes. The government promotes innovation through the establishment and management of publicly owned undertakings, and the number of national research platforms reflects the degree of information flow related to technology and industry in a region and attracts technological innovation talent [53]. The administrative level of a city also affects the flow of factors [54]. As China is a centralised country, cities with high administrative levels concentrate more on social and public resources and play an important role in the intercity flow of international visitors.

### 3.2 Study area

The Yangtze River Delta (YRD) region of China was selected for this empirical study. The YRD region covers an area of 358,000 square kilometres, accounting for 3.7% of the national land area, and its GDP accounts for nearly 25% of the country's total economic output. This region is one of the largest economic zones in mainland China and an important eastern gateway to the world, with a high degree of openness and innovation capability, as well as an important position in the country. According to the Outline of the Yangtze River Delta Regional Integrated Development Plan, the YRD region contains 41 cities (including Shanghai and 40 other cities in Zhejiang, Jiangsu, and Anhui provinces). In this study, we treat the municipal districts, county-level cities, and counties within the scope as independent cities, totalling 195 city units, among which five cities, including Qianshan, Wuwei, Guangde, Hai'an, and Yuhuan, are not identified to the data. Thus, the remaining 190 city units were selected for study (Fig 2).

### 3.3 Data

**3.3.1 Mobile phone data.** We collected cell phone data from China Unicom for a two-month period from 1 September 2019 to 30 October 2019 and examined it against several parameters, including signal emergence time, the latitudinal and longitudinal coordinates of the connected base station, and the user's country of origin. Notably, our dataset does not contain any personal information on the cell phone user, except for the information of the original service provider of the SIM card when set to international roaming. The identification of international visitors and Chinese citizens is quite different. In general, international visitors who

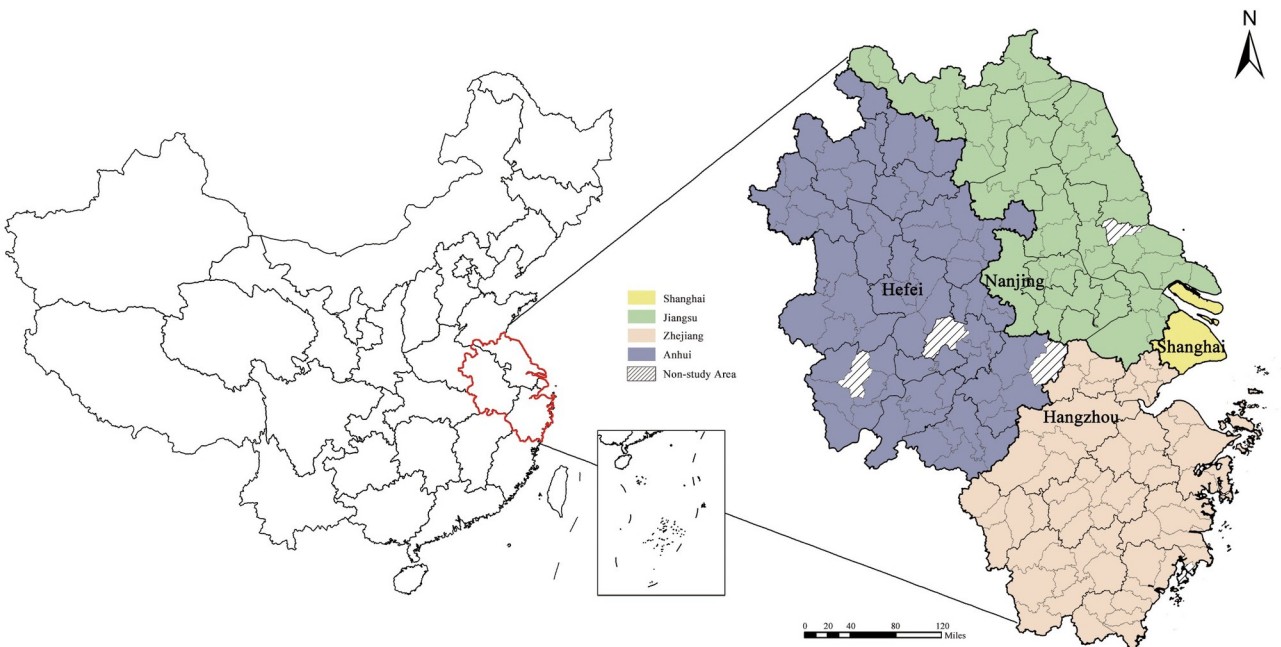

**Fig 2. Map of the study area.** (Source: created by the author based on the base map, the left base map comes from the CIA Factbook (https://www.cia. gov/the-world-factbook/countries/china/map) and the right base map comes from the Open Street Map (OSM) geographic data platform (https://www. openstreetmap.org/copyright)).

have a cell phone operator in their home country are assigned to any Chinese operator based on the source network type and signal strength when they enter China. In other words, any international visitor, that is, the service provider of their cell phone in China, will randomly switch between three communication companies (China Mobile, China Unicom, and China Telecom), depending on their geographical location. In addition, we excluded international visitors connecting these two cities, as well as visitors with mobile country codes from Hong Kong, Macau, and Taiwan. We chose September and October for data collection, which ensured that leisure travel was excluded, as this is a low season for tourism. We believe that cell phone data from China Unicom is a good proxy for the source patterns of foreign visitors.

We collected a total of 3.8 million foreign cell phone users from China Unicom's cell phone set, generating 580,000 users who moved across district and county spatial units and stayed in the unit for more than three hours, including a total of 90,055 cell phone users who stayed for more than one month, generating a total of 2,183,519 intercity movements.

**3.3.2 Impact factors.** The dependent variable in this study was the spatial aggregation of the intercity flows of international visitors, identified in section 3.3.1. Among the independent variables, the enterprises explanatory variable is the selection of the Fortune 500 enterprises in 2020, as well as the collection of the Fortune 500 headquarters enterprises, branches, and mutual investment enterprises of these enterprises in the YRD region, through the China Business Enterprise Registration Database (as of April 2021) (hereafter referred to as "Fortune 500 enterprises"). The database excludes companies with obvious branches, such as banks, insurance, telecommunications, and logistics. Ultimately, 57,456 Fortune 500-related enterprises in operation were included. Air quality (AQI Index or AQI) explanatory variables were obtained from the real-time monitored air quality index on the China Weather website, and daily average AQI data were collected from cities in the study area in 2019 for one year. The total brightness value of nighttime light was extracted from the remote sensing image data of Wuhan

University's "Luojia-1". Other alternative explanatory variables were obtained from the China City Statistical Yearbook in 2019, the Statistical Yearbook of prefecture-level cities, and the public data platforms of provincial governments.

## 3.4 Methods

**3.4.1 Spatial autocorrelation analysis.** Spatial autocorrelation refers to the correlation between a study object and its spatial location, and can be divided into global spatial autocorrelation and local spatial autocorrelation. The correlation study shows that global spatial autocorrelation mainly indicates the correlation degree and overall distribution of attribute values in a certain area space, which can add a necessary explanation to the study [55]. There are many indicators and methods for calculating global spatial autocorrelation, among which the Moran index (Moran's I) is commonly used (Eq (1)):

$$I = \frac{n \sum_{i=1}^{n} \sum_{j \neq i}^{n} w_{ij}(x_i - \bar{x})(x_j - \bar{x})}{\sum_{i=1}^{n} \sum_{j \neq i}^{n} w_{ij} \sum_{i=1}^{n} (x_i - \bar{x})^2} \tag{1}$$

where n represents the number of study objects, $x_i$ and $x_j$ represent the attribute values of the **i**th spatial unit and **j**th spatial unit, respectively, $x$ represents the average attribute value of all spatial units, and $w_{ij}$ is a binary spatial weight matrix used to define the mutual adjacency of spatial units. For Moran's I, the normalised statistic **Z** was used to test the degree of spatial autocorrelation. This can be calculated using Eq (2):

$$Z = \frac{I - E(I)}{\sqrt{VAR(I)}} \tag{2}$$

Moran's I is a global metric of spatial autocorrelation that assumes that the entire space is homogeneous and ignores the instability existing in the space. Considering the need for a more accurate picture of the aggregation characteristics of local spatial elements, the local spatial autocorrelation index of spatial association (LISA) was introduced to calculate the local Moran's I, whose formula is shown in Eq (3):

$$I_i = \frac{(x_i - \bar{x})}{S^2} \sum_{j \neq i}^{n} w_{ij}\left(x_j - \bar{x}\right)$$

$$\text{where } S^2 = \frac{1}{n} \sum_{i=1}^{n} (x_i - \bar{x})^2 \text{ and } \bar{x} = \frac{1}{n} \sum_{i=1}^{n} x_i. \tag{3}$$

**3.4.2 Ordinary least squares (OLS).** Ordinary least squares (OLS) is a statistical method that is commonly used to explain the correlation between a single dependent variable and multiple independent variables [56]. This method was used to analyse the basic regression relationship between the intercity flow of the LSIV and each influencing factor. The corresponding model's equation is as follows:

$$y_i = \beta + \sum_{k=1}^{n} \beta_k x_{ik} + \varepsilon_i \tag{4}$$

where $y_i$ is the dependent variable, $\beta$ is the spatial intercept, $\beta_k$ is the regression coefficient of the independent variable, $x_{ik}$ is the value of the **k**th independent variable in space **i**, **k** is the number of influencing factor variables, and $\varepsilon_i$ is a random error term.

**3.4.3 Geographically weighted regression (GWR).** Previous studies have typically considered the effects of factors on the object of study to be homogeneous, treating all variables as

global or local variables without considering the spatial heterogeneity of influencing factors. Brunsdon (1996) proposed a geographically weighted regression (GWR) analysis, which is a local regression model that extends the classical linear model and is an effective method to study spatial non-stationary data [57]. The method introduces a spatial weight matrix in the linear regression model and geographic coordinates and considers the existence of spatial heterogeneity in the coefficients of independent variables, which can explore the differences in the influence of spatially local geographic location independent variables on the dependent variable. In recent years, this method has been widely applied to the study of local variations in geographical differences [56, 58]. The model structure is as follows:

$$y_i = \beta_0(u_i, v_i) + \sum_k \beta_k(u_i, v_i)x_{ik} + \varepsilon_i \tag{5}$$

where $(u_i, v_i)$ is the spatial coordinate of the $i$th sample point, $\beta_k(u_i,v_i)$ is the value of the continuous function $\beta_k(u,v)$ at point $i$, and $\varepsilon_i$ is the random error of the $i$th region satisfying the spherical perturbation assumptions of mean zero, homoscedasticity, and mutual independence.

The estimated value of the GWR model changes with the spatial weight matrix $w_{ij}$. To avoid estimation error caused by less spatial unit data, it is necessary to choose a weight function to determine $w_{ij}$. In this study, we chose a Gaussian function as the weight function:

$$w_{ij} = \exp\left(-\left(\frac{d_{ij}}{b}\right)^2\right) \tag{6}$$

where $d_{ij}$ is the distance between points $i$ and $j$, and $b$ is the bandwidth. Usually, the larger the bandwidth $b$, the lower the decay rate of the weight influence. Bandwidth is the decisive element in the weight calculation. Therefore, the optimal bandwidth needs to be determined to improve the accuracy of the model.

**3.4.4 Multiscale geographically weighted regression (MGWR).** Considering that the GWR model cannot handle the smoothing problem of each variable in their different spaces, Fotheringham (2017) further proposes the multiscale geographically weighted regression (MGWR) model [59], which takes into account the spatial scale effects of the independent variables on the dependent variable and can satisfy the requirement that each variable uses a specific bandwidth to influence the behaviour of LSIV in space, overcoming the assumption of complete equalisation and complete differentiation of variables, and more closely resembles the real and valid realistic spatial state. The analytical model has recently been applied to the study of influencing factors in a variety of fields, including climate and ecology, transport, science, technology and innovation, and land use etc. [60–63], so this model is equally applicable to this study. This improvement was achieved by reformulating the GWR model as a generalised additive model (GAM) [64]:

$$y = \sum_{j=1}^{k} f_j + \varepsilon_i$$

$$\text{where } f_j = \beta_{bwj}x_j. \tag{7}$$

where $f$ is the smoothing function applied to the $j$th explanatory variable, which can be characterised by different bandwidth parameters, and $\varepsilon_i$ is the random error. Each regression coefficient in the MGWR is based on a local regression and has a different bandwidth setting, whereas in the classical GWR, all variables have the same bandwidth.

**3.4.5 Cluster analysis.** Cluster analysis is a multivariate statistical analysis method that divides a sample into multiple categories comprising similar objects. Cluster analysis allows

for the exploration of homogeneity within groups, correlations, and major differences across groups [65]. K-means clustering methods perform clustering through continuous iterations until the desired result is achieved. We use the squared Euclidean distance to describe the similarity, as shown in Eq (8):

$$D = \sqrt{\sum_{i=1}^{n} (x_{ik} - y_{jk})^2} \tag{8}$$

where **D** is the Euclidean distance, **i** is the number of samples, and **k** is the number of indicators.

## 4. Results

### 4.1 Spatial patterns of LSIV

Aggregating the intercity daily flows of LSIV in the YRD region by city units, the spatial pattern shows a clear core-periphery structure, with higher international visitor arrivals in the eastern coastal and along the Yangtze River regions, and significantly lower arrivals in the western, northern, and southern regions of the YRD. At the same time, the spatial distribution of the LSIV scale shows a hierarchical distribution. Because the numerical spatial distribution shows a head-tail distribution (Fig 3), stratification was performed using Jiang's head-tail distribution method [66]. This shows that Shanghai has much more international long-term visitors than other cities and is classified as the first tier. Hangzhou, Nanjing, and Suzhou, three cities with higher internationalisation, are classified as the second tier. Wuxi and Ningbo, two cities with developed manufacturing industries, are third tier cities. Hefei, Changzhou, Yangzhou, Wenzhou, Nantong, and Kunshan are in the fourth tier, while the remaining cities are in the fifth tier.

The spatial correlation trend of the inflow and outflow of the LSIV in the study area is fundamental for subsequent modelling. Therefore, we conducted a global Moran's I analysis of the international visitor volumes in the study area. After normalising the variance, Moran's I ranges from –1.0 to +1.0. When Moran's I is greater than 0, the data show a positive spatial correlation, and the larger Moran's I, the more obvious the spatial correlation; when Moran's I is less than 0, the data show a negative spatial correlation, and the smaller the spatial variation, the larger Moran's I. When Moran's I is less than 0, the data show a negative spatial correlation, and the smaller the spatial variation, the larger Moran's I. The experimental results were considered significant when the p-value was less than 0.05 and the Z value was greater than 1.96. Moran's I is 0.0497 and the Z value is 2.8548, which indicates that the travel destinations of international visitors have spatial autocorrelation characteristics and should be analysed for spatial heterogeneity.

We used Anselin Local Moran's I (LISA) to explore the spatial autocorrelation of international visitors' travel destinations (Fig 4). Shanghai forms a high-high cluster with Suzhou, Wuxi, Kunshan, and Shaoxing municipal districts, which is a high-low cluster, indicating that Xuzhou has a high inflow and outflow of international visitors and shows statistically significant high and low value differences with neighbouring counties. Shanghai, Nanjing, and Hangzhou municipal districts adjacent to Taicang, Liyang, Langxi, Jiashan, Pinghu, Haining, Tongxiang, and Deqing counties, and the cities form low-high clusters, indicating that the surrounding mega-cities present statistically significant low and high value differences. Parts of northern Jiangsu, southern Zhejiang, and western Anhui are low-low clusters of international visitor inflows and outflows.

### 4.2 Results of OLS, GWR, and MGWR models

**4.2.1 Exploratory regression model.** To eliminate differences in numerical magnitude, we standardised all explanatory variables and screened indicator combinations based on the

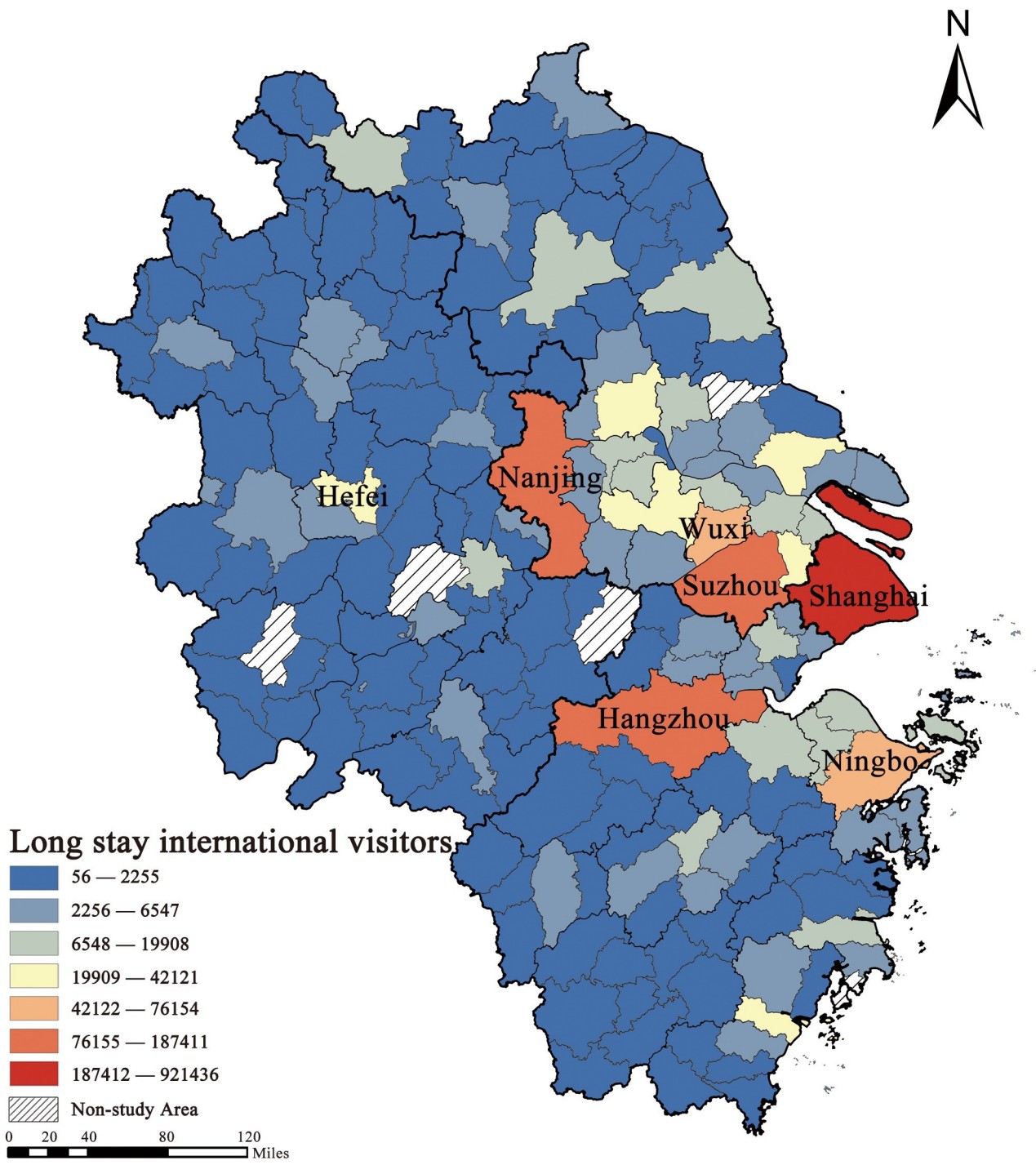

**Fig 3. LSIV's spatial distribution.** (Source: created by the author based on the base map which comes from the Open Street Map (OSM) geographic data platform (https://www.openstreetmap.org/copyright)).

following principles: (i) the combination of indicators obtained a large corrected $R^2$ ($R^2 > 0.45$), (ii) each indicator was as significant as possible at the 10% level ($p<0.1$), (iii) the VIF value was well below 7.5, and (iv) the Akaike information criterion (AIC) for the selected combination of indicator values for the selected combination of indicators was as small as possible.

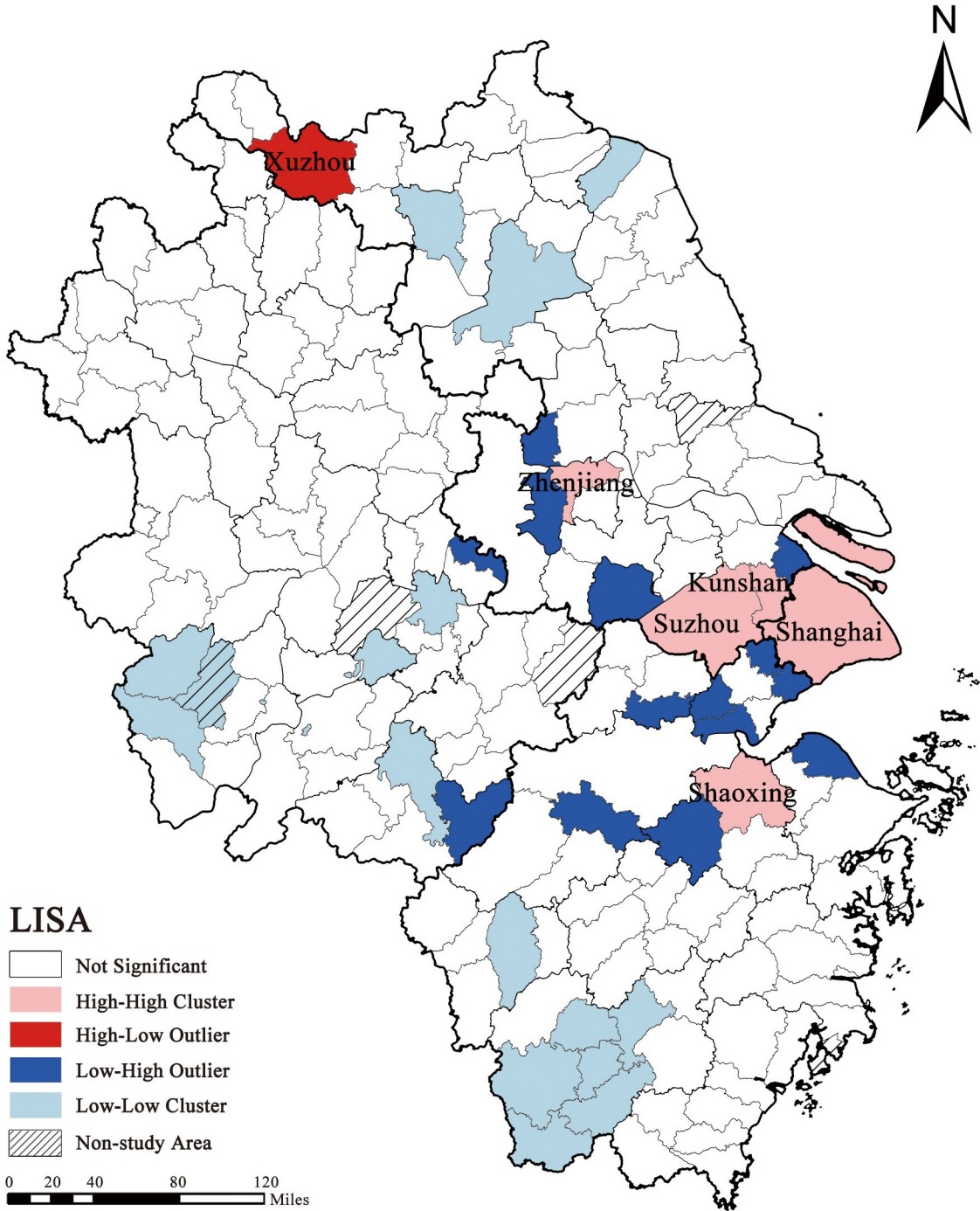

**Fig 4. LISA of LSIV map.** (Source: created by the author based on the base map which comes from the Open Street Map (OSM) geographic data platform (https://www.openstreetmap.org/copyright)).

Combined models were built with the help of the ArcGIS exploratory regression tool. The exploratory regression tool simulates thousands of possible models from a complete list of predictor variables and then outputs the best model based on the adjusted $R^2$ and AICc, among other goodness-of-fit criteria. Using the exploratory regression tool, a combination model with five independent variables was selected because using more variables did not significantly

**Table 2. Exploratory regression analysis results.**

| Adjusted R² | AICc | MaxVIF | Models | | | | | | |
|---|---|---|---|---|---|---|---|---|---|
| 0.9872 | -279 | 7.44 | +enterprises*** | +AQI* | -income** | +imports and exports*** | -administrative hierarchy*** | | |
| 0.9868 | -274 | 7.22 | +enterprises*** | -income*** | +imports and exports*** | -administrative hierarchy*** | | | |
| 0.9884 | -269 | 7.06 | +enterprises*** | +AQI** | +imports and exports*** | -administrative hierarchy*** | | | |
| 0.9859 | -263 | 6.90 | +enterprises*** | +imports and exports*** | -administrative hierarchy*** | | | | |
| 0.9854 | -256 | 6.51 | +enterprises*** | -income*** | +imports and exports*** | | | | |
| 0.9849 | -250 | 7.27 | +enterprises*** | +imports and exports*** | -nighttime light*** | | | | |
| 0.9843 | -242 | 2.85 | +scenic spots*** | +enterprises*** | -scientific research platforms*** | | | | |
| 0.9818 | -213 | 7.46 | -AQI** | +foreign capital*** | -income*** | +imports and exports*** | -transportation systems** | | |
| 0.9651 | -89 | 2.48 | -AQI** | +foreign capital*** | -income* | -scientific research platforms*** | -transportation systems** | | |
| 0.9618 | -69 | 6.24 | +GDP*** | -scenic spots* | -income*** | -scientific research platforms*** | -transportation systems* | -administrative hierarchy*** | -nighttime light*** |
| 0.9612 | -67 | 6.22 | +GDP*** | -scenic spots** | -income*** | -scientific research platforms*** | | -administrative hierarchy*** | -nighttime light*** |
| 0.9607 | -65 | 6.09 | +GDP*** | | -income*** | -scientific research platforms*** | -transportation systems* | -administrative hierarchy*** | -nighttime light*** |

*$p < 0.05$

**$p < 0.01$

***$p < 0.001$.

improve the adjusted $R^2$ or AICc (Table 2). The output variable combination models were tested using OLS, GWR, and MGWR, and the best-performing MGWR model was finally selected. This five-variable combination model includes the following predictor variables: enterprises, AQI, income, imports and exports, and administrative hierarchy [67].

**4.2.2 Comparison of OLS, GWR, and MGWR models.** The GWR and MGWR models significantly improved the explanatory power of the model. The residual sum of squares gradually decreased from OLS > GWR > MGWR, while the AICc value gradually increased. Overall, the results indicated that the MGWR model was the best choice.

International visitor long-term travel volume was found to be characterised by spatial correlation and spatial non-stationarity. In this context, traditional linear OLS regression models have difficulty in producing satisfactory results, and the GWR and MGWR models were used to explore the factors of international visitor long-term travel volume. The GWR and MGWR models can be used to explore spatial variation at a certain scale by creating local regressions at each point in the spatial range and driving factors. Theoretically, GWR and MGWR have higher accuracies than traditional linear regression models. As shown in Table 3, by introducing heterogeneity, MGWR and GWR improved model performance in the OLS method. The goodness-of-fit of GWR was 0.997, while that of MGWR was 0.998. Additionally, the residual sum of squares gradually decreased, the sigma value gradually decreased, and the AICc value gradually decreased from the OLS method to the GWR model and then the MGWR model. In general, the residual sum of squares and AICc values are auxiliary measures of

**Table 3. Overall performance of all models.**

| Diagnostic metric | OLS | GWR | MGWR |
|---|---|---|---|
| AICc | -279.006 | -566.200 | -608.293 |
| Adjusted $R^2$ | 0.987 | 0.997 | 0.998 |
| Residual sum of squares | 2.372 | 0.323 | 0.288 |
| sigma | 0.114 | 0.048 | 0.043 |

model accuracy; the smaller the residual sum of squares and AICc values of the model, the better the model fit.

**4.2.3 Scale analysis of the MGWR model.** The MGWR model is an improved version of the GWR model, and the bandwidths of different variables in MGWR are different. The MGWR model can directly reflect the differential action scales of different variables, whereas the classical GWR model can only reflect the average of the action scales of individual variables. Table 4 shows that the bandwidth of the classical GWR is 52. However, by calculating the MGWR model, the action scales of the different variables were found to be very different. In the MGWR regression results, the action scales of income, imports and exports were 188 and 162, respectively, which are close to the global scale, indicating that the spatial distribution of these two factors is relatively stable and there is almost no spatial heterogeneity. In other words, no matter where in the study area, the same change in long-term visitor travel volume was observed, considering the influence of these factors. The scales of 43, 43, and 45 for enterprises, AQI, and administrative hierarchy, respectively, were small, indicating a high degree of spatial heterogeneity and variability across regions in the effects of these variables on changes in long-term visitor travel. Here, the intercept indicates the effect of different locations on LSIV trips when other independent variables are determined.

## 4.3 Analysis of the impact mechanism

The statistical results and spatial distributions of the coefficients in the MGWR are presented in Table 5. The signs of the regression coefficients indicate the corresponding enhancement or suppression of the size appreciation of the LSIV, whereas the absolute values indicate the strength of the impact. In general, the direction and strength of the regression coefficients of the listed variables were different. Enterprises had the largest standard deviation among the tested variables, while income, imports, and exports had the smallest standard deviation. These results indicate that there is a consensus among the residents of the study area on these two variables.

Interestingly, the most significant difference in the effect of city enterprises on LSIV is found in the regression coefficient between 0.180 and 0.881. This relationship shows that, in one city, when enterprises increased by 1%, the number of LSIV increased by 18.0%, while in

**Table 4. MGWR model bandwidth.**

| GWR | MGWR |
|---|---|
| 52 | 55(Intercept) |
| | 43(enterprises) |
| | 43(AQI) |
| | 188(income) |
| | 162(imports and exports) |
| | 45(administrative hierarchy) |

**Table 5. Summary statistics for MGWR parameter estimates.**

| Variable | Mean | STD | Min | Median | Max |
|---|---|---|---|---|---|
| Intercept | -0.026 | 0.021 | -0.068 | -0.023 | 0.013 |
| enterprises | 0.527 | 0.202 | 0.180 | 0.522 | 0.881 |
| AQI | 0.013 | 0.024 | -0.016 | 0.005 | 0.090 |
| income | -0.000 | 0.002 | -0.003 | -0.001 | 0.003 |
| imports and exports | 0.177 | 0.004 | 0.173 | 0.176 | 0.190 |
| administrative hierarchy | 0.005 | 0.026 | -0.060 | 0.011 | 0.051 |

another city, when enterprises increased by 1%, the number of LSIV increased by 88. 1%. Explanations for these results include that the number of Fortune 500 companies and their partner companies within the YRD region varied widely, and that international visitors stayed in the YRD region for long-term purposes more for work, such as business travel, an essential difference from the short-stay international visitors, whose main focus was travel. The LSIV also paid more attention to the administrative level of the city, and the administrative system characteristics of Chinese cities also have an impact on international visitors. Cities with high administrative levels have easier access to resources, better economic development, and higher levels of urban construction. Thus, they are more likely to be favoured by international visitors. Residents' incomes and the total import and export of cities did not have a significant impact on the intercity daily flow of LSIV, which is somewhat different from findings reported in the literature [68].

The coefficient of determination (adjusted $R^2$) of the MGWR results for different variables varied considerably. Fig 5A shows the distribution of several selected correlation coefficients in the study area. Cities enterprises were significantly and positively correlated with LSIV arrivals (Fig 5B). Specifically, the positive effect of the number of companies on LSIV arrivals gradually increased from west to east in the YRD region, with a circling distribution and an average regression coefficient of 0.951. The closer the location was to the southeast of the delta region, the greater the number of Fortune 500 companies in the city, the stronger the attraction to LSIV, and thus the larger the size of the international visitor attraction. Surprisingly, the spatial scope of the greatest influence of the number of enterprises on the attraction of LSIV was found to be exactly the same as the planning scope of the Shanghai Metropolitan Area. This indicates that the scope of the Shanghai Metropolitan Area was the concentration area of foreign companies in the YRD region, which also attracts foreign visitors strongly. Second, the influence of enterprises in the northern and central regions of Zhejiang Province on the attraction of LSIV was better than that of Jiangsu Province, which has a stronger economic strength.

The impact of AQI on the LSIV arrivals showed positive and negative differences, with a weak negative spatial correlation in most of Anhui, Nanjing, Jiangsu, and Ning and Zhoushan in the eastern Zhejiang Province (Fig 5C). The strength of the effect of air quality on LSIV arrivals increased from west to east, with Shanghai and the surrounding areas being the most influential regions, with an average regression coefficient of 0.013 and less variation overall. The importance of air quality for residents of Western countries leads to travel choices that are easily influenced by the air quality of Chinese cities [69]. At the same time, as Chinese residents pay more attention to the health and ecological environment, their intercity travel is also negatively affected by air pollution to some extent [70]. However, we found that the intercity travel of LSIV in the YRD region was less affected by AQI, which may be related to the recent significant improvement in air quality in the YRD region of China.

Income and LSIV were only spatially positively correlated in the northwestern region and negatively correlated in all the other cities in the YRD region (Fig 5D). Income was close to the

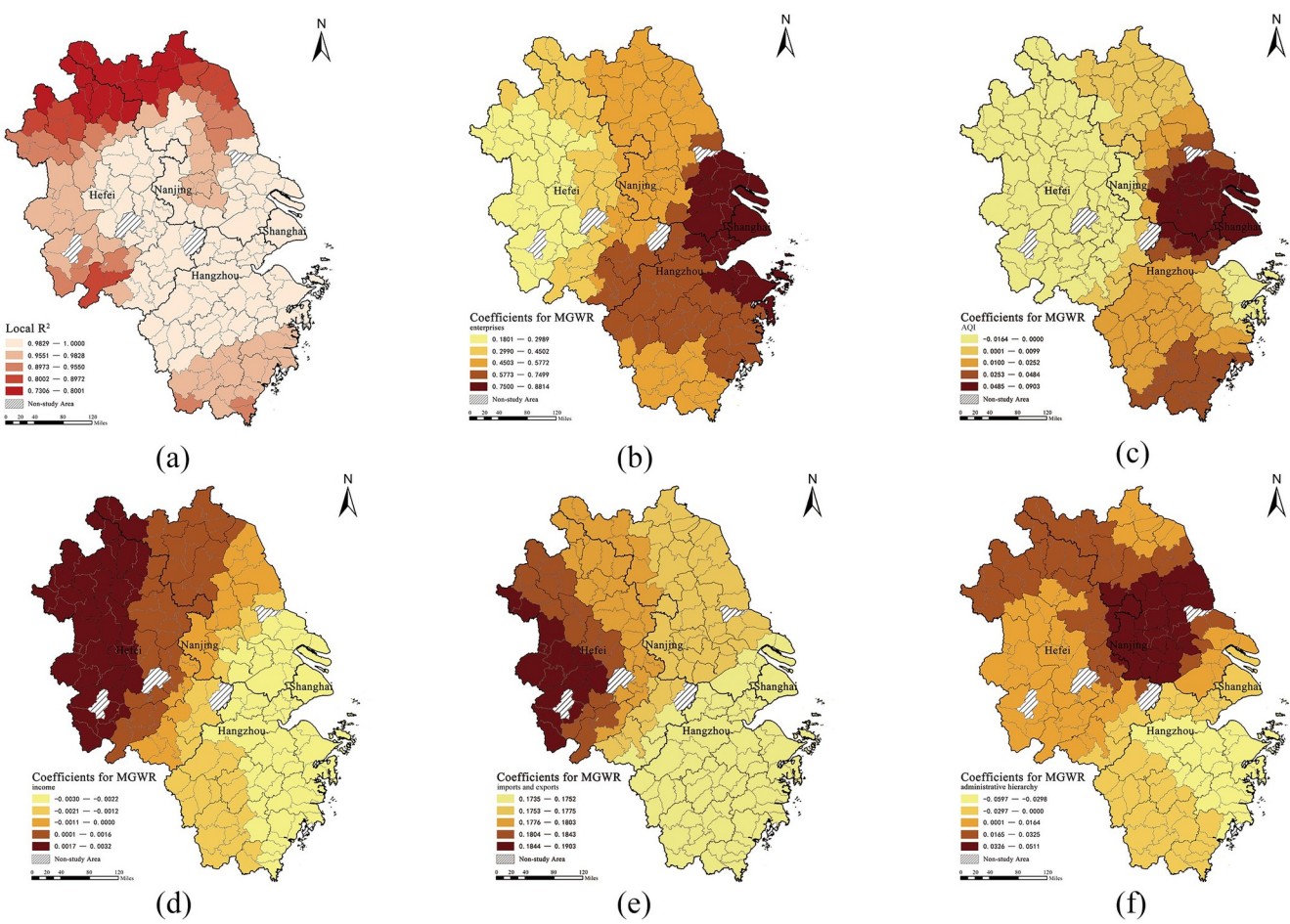

**Fig 5. Spatial patterns of coefficients in the MGWR.** (Source: created by the author based on the base map which comes from the Open Street Map (OSM) geographic data platform (https://www.openstreetmap.org/copyright)).

global variable, with little variation in the spatial action scale. In the context of the entire study region, the effect of residents' income levels on LSIV was not significant. Previous studies have generally concluded that urban and rural residents' income has a significant effect on the intercity travel of local residents [68]. However, in this study, we found that the overall income level of urban and rural residents did not have a direct or significant impact on the arrival of long-term international visitors.

The exports and imports of cities are significantly and positively correlated with LSIV arrival (Fig 5E). Exports and imports were weakly positively correlated with international visitor arrivals, and this variable was close to the global variable with little difference in the spatial action scale. From the perspective of the entire YRD region, the influence of imports and exports on LSIV was relatively stronger in western cities, and the general trend was that this influence gradually decreased from northwest to southeast, which may be due to the smaller total foreign investment in the western YRD region, which is more sensitive to data changes.

The administrative hierarchy of cities was positively correlated with LSIV arrivals in the northern part of the YRD region and negatively correlated in the southern part of the YRD region (Fig 5F). The general trend was that this effect gradually weakened in the central region, consisting of Nanjing, Zhenjiang, Yangzhou, Wuxi, Changzhou, Taizhou, and Yancheng, towards the periphery, showing a circular spatial distribution. However, the traditional

**Table 6. Average clustering results for the 4 types of cities.**

| | Category | | | |
|---|---|---|---|---|
| | **A** | **B** | **C** | **D** |
| Number of cities | 28 | 37 | 83 | 42 |
| Proportion of cities | 14.74% | 19.47% | 43.68% | 22.11% |
| enterprises | 0.867596 | 0.655435 | 0.497817 | 0.243223 |
| AQI | 0.04408 | 0.019409 | 0.009471 | -0.00524 |
| income | -0.00278 | -0.00216 | -0.0003 | 0.002228 |
| imports and exports | 0.174847 | 0.174363 | 0.176538 | 0.183187 |
| administrative hierarchy | -0.00658 | -0.02518 | 0.018348 | 0.013594 |
| Impact level Low | | | | High |

perception is that intercity mobility should be clustered in large cities with high administrative levels. To our surprise, the LSIV in the YRD region was not influenced by the administrative levels of cities, which may be related to the fact that international visitors generally reside in high-ranking cities.

## 4.4 Cluster analysis

We categorised the coefficients affecting the magnitude of the intercity flow of the LSIV to identify trends across the study area. A systematic clustering method based on intraclass squared Euclidean distance was used to classify the 190 cities in the study area into four types, for which the mean of all variables for each type were calculated (Table 6). The impact of each indicator coefficient on the number of long-stay visitors was measured based on the absolute value of the impact. Finally, the images were visualised on a map (Fig 6).

Overall, the spatial distribution of cities in the abovementioned categories was more pronounced, and the effects of different variables on appreciation rates reflected the preference for international visitor travel in the region over a certain period of time. Category A include Shanghai and its neighbouring cities, Suzhou, Wuxi, Nantong, Jiaxing, Ningbo, and Zhoushan. It is worth noting that this range was exactly the same as that of the Greater Shanghai Metropolitan Area Collaborative Spatial Planning released by the central government in 2022, suggesting that there are common characteristics in the attraction of international visitor trips by the relevant attributes of cities in this range. Enterprises, AQI, and income were the greatest attractions for international visitors in the YRD region, the impact of imports and exports was at a medium level at the regional scale, and the impact factor of the administrative hierarchy was at the lowest level. The scope of the Shanghai Metropolitan Area is the area with the highest outwards economic development in the YRD region, with more foreign enterprises related to Fortune 500, the area with the most rapid urban economic development in the past 40 years, the highest residents' incomes, and the best management of air quality. Therefore, cities in the Shanghai metropolitan area are attractive to LSIV, but are not sensitive to the administrative hierarchy of the city.

Categories B and C included the middle regions of the Yangtze River Delta, mainly Zhejiang province, the vast majority of cities in Jiangsu Province, and the southern part of Anhui province. The overall economic development level of these regions, except for a few provincial capital cities, was relatively lower than that of Type A regions. Enterprises, AQI, and imports and exports were at the middle level of the region in terms of attracting international visitors to travel, and the impact of income was at a lower level. However, the attractiveness of the administrative hierarchy to international visitors in the region was above average, indicating

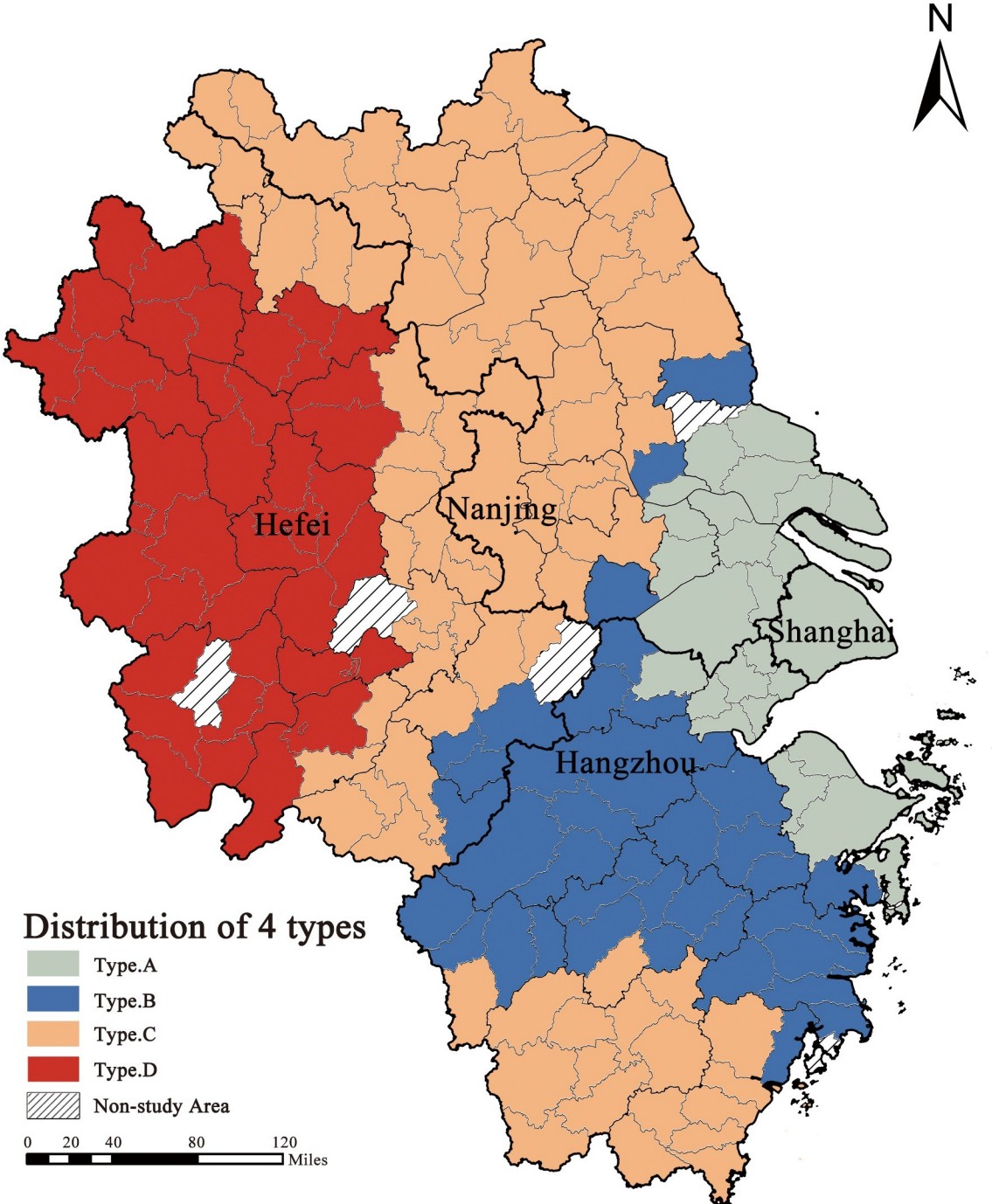

**Fig 6. The spatial distribution of four types of cities.** (Source: created by the author based on the base map which comes from the Open Street Map (OSM) geographic data platform (https://www.openstreetmap.org/copyright)).

the concentration of resources, such as public service facilities, in the capital cities of these regions, while the periphery was relatively weak, resulting in more attractive capital cities. Therefore, more cities in the central part of the YRD region need to allocate better public service facilities to attract more foreign visitors and create more business and investment opportunities.

Category D includes the western part of the YRD region, which is mainly comprised of cities in Anhui province. These areas traditionally lag behind in economic development in the YRD region, and the impact of various factors on the appreciation rate is at a relatively low level.

## 5. Conclusion and discussion

### 5.1 Conclusion

To address the shortcomings of past statistical or survey data in capturing the intercity daily flow of long stay international visitors (LSIV), this study examines the intercity daily flow of international visitors using mobile phone datasets of foreign subscribers identified by China Unicom, capturing a large amount of mobile location-based big data on international visitor activities, thereby providing a novel research perspective. To be precise, based on the construction of an "attribute-network" theoretical analysis framework, mobile phone data from foreign cell phone users who stayed in the YRD region for more than one month during September and October 2019 were used to map the spatial patterns of intercity mobility and analyse the related socio-economic and other influencing factors. The MGWR model results showed that the intercity mobility of LSIV was significantly associated with enterprises and weakly associated with AQI and administrative hierarchy, with different cities within the YRD region of China having different natures and strengths of association with these factors. By classifying the various coefficients that affect the scale of intercity mobility of LSIV, the YRD region can be classified into four types, and urban planning strategies can be tailored accordingly.

Based on the construction of an "attribute-network" theoretical analysis framework, this study uses mobile phone data from foreign cell phone users who stayed in the YRD region for more than one month during September and October 2019 were used to map the spatial patterns of intercity flow, and analyses the spatial heterogeneity of related socio-economic and other influencing factors using the MGWR model. heterogeneity, and the resulting coefficients were analysed by clustering. The following conclusions are drawn:

**(1) Spatial patterns of LSIV.**　Visitor arrivals in Shanghai are much higher than those in other areas of the YRD. Overall, LSIV arrivals are high in the eastern coast and along the Yangtze River in the YRD, and lower in the southern and northern areas, presenting a clear core-edge structure. Through local spatial autocorrelation analysis, it is found that the periphery of the megacities exhibits the difference between significantly high values in the megacities and low values in the peripheral districts and counties.

**(2) Spatial heterogeneity of impact factors.**　In terms of the scale of action, the spatial distributions of income, imports and exports are more stable, with little spatial heterogeneity. The scales for enterprises, AQI and administrative hierarchy are all smaller, indicating a high degree of spatial heterogeneity and variability across regions in the impact of these variables on changes in long-term visitor travel. In terms of impact benefits, enterprises, AQI, imports and exports all have a positive effect on LSIV stays, while there are positive and negative differences in income and administrative hierarchy, with income having a positive effect in the north-west and a negative effect elsewhere, and administrative hierarchy having a positive effect in the north and a negative effect in the south. In terms of the intensity of impacts, the intensity of impacts is high for enterprises, imports and exports, where the spatial variation in the impact of firms on LSIV is also high, while the intensity of impacts is generally low for income, AQI, and administrative hierarchy.

**(3) Spatial clustering of impact factors.**　The clustering of the coefficients divides the study area into four categories, with the scope of category A being exactly the same as the planning scope of the Shanghai Metropolitan Area, where the key influences on LSIV stays are

enterprises, AQI, and income, while administrative hierarchy has little effect on them, suggesting that the development of the area is characterised by a flattening of the area[6]. Categories B and C include the central Yangtze River Delta region, where the key influencing factor is the administrative hierarchy, which is a result of the region's greater concentration of resources in provincial capitals with high administrative hierarchie [71]. Category D includes the western part of the YRD region, where the key influencing factor is the imports and exports, mainly because the region is economically backward and more sensitive to changes in investment figures.

## 5.2 Discussion

As a special flow of people across great distances, the extended stay of international visitors in a country or region brings potential benefits of knowledge innovation and economic growth to the destination [3]. However, it has long been the main line of research to focus on transnational human mobility based on the Population Migration theory [5, 6]. Our research has found that the intercity daily flow of LSIV within a country and region not only has the travel characteristics of residents in the country of origin, but also the life process gradually receives the influence of intercity daily flow of local residents. Understanding the spatial characteristics and influencing factors of the intercity daily flow of LSIV based on the Space of Flow theory links individual travel behaviour at the micro-level with the spatial patterns of intercity connections at the macro-level, which is a novel perspective.

Our results uncover some interesting phenomena: firstly Population Migration theory suggests that the income level of a destination significantly affects the movement of people, with populations tending to flow from areas with low income levels to areas with high income levels, but our findings find that local income levels do not have a direct and significant impact on the arrival stays of long-term international visitors [68, 72]. Secondly, while previous studies have argued that due to the siphoning phenomenon, cities with high administrative levels attract more people to visit and stay, our results find that LSIV is not affected by the administrative level of the city [71]. Therefore, it can be assumed that the traditional "push" and "pull" factors do not play a significant role in our research object, which confirms the inapplicability of the "push-pull" model to this study. This also confirms that the "push-pull" model is not applicable to this study.

In addition, this study provides insights into the relationship between urban attributes and urban networks. Recent research has focused on the perception of urban network patterns from the Space of Flow theory, ignoring the equally important role that urban attributes play in this process [73]. Specifically, any space of flow needs to exist as a spatial carrier with space of place, and the socio-economic and other attribute characteristics of space of place will influence human daily activities. In this context, this study evaluates for the first time how the daily flow of LSIV is influenced by differences in city attributes by constructing a multiscale geographically weighted regression model.

Using data with timely and fine spatial resolution, this study goes beyond earlier studies by capturing the relative attractiveness and spatial heterogeneity in the intercity flow of the LSIV within the YRD region of China. The results of the MGWR model suggest that LSIV in the YRD region tend to cluster in cities with a high number of Fortune 500 enterprises, good air quality, and a high administrative ranking. Residents' incomes and cities' imports and exports did not play a significant role. Our results indicate that the dynamics of intercity flow of LSIV are not identical to those of intercity flow of local residents [9], nor are they the same as those of international visitors who travel for short periods of time [74]. This study also demonstrates that LSIV in China are adapting to the socio-economic and built environment of Chinese cities while retaining their long-established travel habits.

Long stay international visitors are an important manifestation of regional integration in economic globalisation. This study finds that the eastern economically developed regions of the YRD region are more attractive to LSIV overall, with the Shanghai Metropolitan Area being the most attractive, notably far more attractive than other regions, with obvious spatial differences. The study illustrates the marked spatial imbalance in the intercity flow of LSIV, reflecting the "unbalanced and insufficient" socio-economic development of the YRD region. In recent years, China has made efforts to continue to expand its national development strategy of opening up to the outside world in the hopes of achieving common prosperity through balanced regional development. Our findings indicate that the YRD region should continue to strengthen urban planning and construction to attract investments from multinational enterprises, improve the living environment, and upgrade the resources of provincial capitals to concentrate public service facilities.

Although the current study has made some interesting findings from new theoretical perspectives and has proposed a development path for the future of the YRD, there are some limitations to this study and further in-depth research is needed to support the refinement of our ideas. The mobility of people travelling is a complex institutional process that is influenced by a combination of macro-environmental and individual decision-making [13]. Our current study discusses the influences of LSIV at the macro-level of the built environment, i.e., in terms of how city attributes attract visitors, but there is no way to know how visitors' own intentions affect their arrival and stay. The next phase of our research will be a questionnaire survey of international visitors in the YRD region of China to explore the impact of individual decision-making intentions on city arrival stays in terms of individual information background, travel purpose, travel preferences, cultural experiences and social interactions etc.

## Supporting information

**S1 File. Statistical data of all counties and cities in the Yangtze River Delta in 2019.**
(ZIP)

## Acknowledgments

We would like to thank the National Platform for Common Geospatial Information Services for providing open source maps (https://www.tianditu.gov.cn/).

## Author Contributions

**Conceptualization:** Yao Wang, Xiaodong Meng.

**Data curation:** Yao Wang, Meilin Zhu.

**Formal analysis:** Meilin Zhu.

**Funding acquisition:** Yao Wang.

**Methodology:** Yao Wang, Meilin Zhu, Xiaodong Meng.

**Software:** Meilin Zhu.

**Writing – original draft:** Yao Wang, Meilin Zhu, Xiaodong Meng.

**Writing – review & editing:** Yao Wang, Meilin Zhu, Xiaodong Meng.

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
