## [Decision Letter · Decision Letter 0]

1 Jun 2023

PONE-D-23-12368Spatial Impacts of the Intercity Flow of Long Stay International Visitors Based on Mobile Phone Data in Yangtze River Delta, ChinaPLOS ONE

Dear  Dr. MENG,

Thank you for submitting your manuscript to PLOS ONE. After careful consideration, we feel that it has merit but does not fully meet PLOS ONE’s publication criteria as it currently stands. Therefore, we invite you to submit a revised version of the manuscript that addresses the points raised during the review process.

We look forward to receiving your revised manuscript.

Kind regards,

Najabat Ali, Post doc

Academic Editor

PLOS ONE

Journal Requirements:

7. We note that Figures 2,3,4,5 and 6 in your submission contain [map/satellite] images which may be copyrighted. All PLOS content is published under the Creative Commons Attribution License (CC BY 4.0), which means that the manuscript, images, and Supporting Information files will be freely available online, and any third party is permitted to access, download, copy, distribute, and use these materials in any way, even commercially, with proper attribution. For these reasons, we cannot publish previously copyrighted maps or satellite images created using proprietary data, such as Google software (Google Maps, Street View, and Earth). For more information, see our copyright guidelines: http://journals.plos.org/plosone/s/licenses-and-copyright.

a. You may seek permission from the original copyright holder of Figures 2,3,4,5 and 6 publish the content specifically under the CC BY 4.0 license.  

Reviewers' comments:

Reviewer's Responses to Questions

**Comments to the Author**

1. Is the manuscript technically sound, and do the data support the conclusions?

Reviewer #1: Yes

Reviewer #2: Yes

2. Has the statistical analysis been performed appropriately and rigorously? 

Reviewer #1: Yes

Reviewer #2: Yes

3. Have the authors made all data underlying the findings in their manuscript fully available?

Reviewer #1: Yes

Reviewer #2: Yes

4. Is the manuscript presented in an intelligible fashion and written in standard English?

Reviewer #1: Yes

Reviewer #2: Yes

5. Review Comments to the Author

Reviewer #1: Title: Spatial Impacts of the Intercity Flow of Long Stay International Visitors Based on Mobile Phone Data in Yangtze River Delta, China

The article titled " Spatial Impacts of the Intercity Flow of Long Stay International Visitors Based on Mobile Phone Data in Yangtze River Delta, China" addresses the spatial impacts of intercity flow of long stay foreign visitors on mobile data. discussed spatial dependence using multiscale geographically weighted regression (MGWR), and performed cluster analysis to understand the combination effects for the Yangtze River Delta (YRD) region in 2019. This study unfolds innovative insights, and it is comprehensive in its nature. However, it needs a few improvements to be accepted for publication. Hopefully, my detailed comments will help the authors to improve the manuscript's quality.

1. The problem definition and innovations presented in the introduction must be greater clarity and specificity. Rewrite the study’s contribution in sequence.

2. Literature review section is thin and have cited old references. Conduct a rigorous effort to update the literature review section by adding updated references from recent studies.

3. Methods are advanced, appropriate, and accurate. However, add some detail to justify the application of these techniques in this study.

4. Be consistent with table headings. Recheck the table numbers.

5. Add more citations in the explanations of results in discussion, also discuss the results more appropriately.

6. Revise the conclusion to describe your study's problem, methods, and innovative findings. Add limitations of the study.

7. Revise for typo mistakes.

Reviewer #2: It is my pleasure to review the manuscript for the esteemed journal. After reading this manuscript, i am suggestion some minor changes that may further improve this manuscript.

Abstract: Abstract section is relatively lengthy ,it should be short and to the point.

Introduction ,Literature review and methodology section is fine and very well written.

Result,discussion and conclusion : The arrangement of discussion and conclusion is not as per requirement of journal. Author first conclude the paper than include the discussion section, and this discussion section is also missing the support of previous studies.

6. PLOS authors have the option to publish the peer review history of their article (what does this mean?). If published, this will include your full peer review and any attached files.

Reviewer #1: No

Reviewer #2: No

---

## [Author Response · Author response to Decision Letter 0]

26 Jul 2023

Comments from the reviewers:

Reviewer #1:

The article titled " Spatial Impacts of the Intercity Flow of Long Stay International Visitors Based on Mobile Phone Data in Yangtze River Delta, China" addresses the spatial impacts of intercity flow of long stay foreign visitors on mobile data. discussed spatial dependence using multiscale geographically weighted regression (MGWR), and performed cluster analysis to understand the combination effects for the Yangtze River Delta (YRD) region in 2019. This study unfolds innovative insights, and it is comprehensive in its nature. However, it needs a few improvements to be accepted for publication. Hopefully, my detailed comments will help the authors to improve the manuscript's quality.

1. The problem definition and innovations presented in the introduction must be greater clarity and specificity. Rewrite the study’s contribution in sequence.

Response: Thank the experts for their affirmation of the research in this article, and pointing out the relevant shortcomings. We have made the relevant changes in this section taking full account of your suggestions.

Firstly, with regard to the problem definition aspect, we have been more specific in describing and defining long stay international visitors(LSIV), LSIV in this paper refers to foreign visitors who stay in the destination country for more than 1 month, and we have highlighted the mechanisms and characteristics of their travel behavior. 

Secondly, concerning the innovation section of the study, we have added more details to the research gaps and innovations in research methods based on the original manuscript that has already described the innovations in identifying data of long stay international visitors. Most of the current studies analyze the human mobility from the perspectives of both external representations and internal causes, but the role played by urban attributes in the mobility process and the explanation of spatial heterogeneity of mobility The attention is less on both, therefore, our study attempts to build reveal the differences in the spatial patterns of daily inter-city movements of long stay international visitors, explain the socio-economic factors associated with mobility patterns, and fill the relevant research gaps. Meanwhile, we use a multiscale geographically weighted regression model(MGWR) to explore the spatial heterogeneity of influencing factors, which further considers the differences in the bandwidth of action of each variable based on the original analysis method, which will be explained in detail in the methods section of Chapter 3. 

Finally, regarding the contribution of this paper, we have presented it in the order of the chapters in the article, but we have taken your suggestions fully into account and have provided a more detailed sub-section of the contribution section. The main areas include: (1) contribution to the theoretical framework; (2) innovation in research methodology; (3) significance of spatial heterogeneity results; and (4) significance of spatial clustering.

2. Literature review section is thin and have cited old references. Conduct a rigorous effort to update the literature review section by adding updated references from recent studies.

Response: Thanks for this suggestion. Firstly, In response to the suggestion that "Literature review section is thin", we have made major global changes to the literature review section and divided it into the following three subsections: (1) In section 2.1 Population Migration theory and Space of Flow theory, we have retained the previous introduction to the Space of Flow theory and added a section on Population Migration theory. We also add a section summarizing the characteristics, similarities, and differences of the two theories. (2) In Section 2.2 Residential migration and human flow, we retain the previous section on the mechanisms and implications of human mobility, add an explanation of the motivation of human migration, discuss the effects and implications of these two behaviors, and address the spatial differences between residential migration and human flow. (3) 2.3 Impact factors of residential migration and human flow. In this section, we retain the analysis of factors influencing human flow, add a discussion of factors influencing resident migration under the "push-pull" theory, and compare the influencing factors of both, and find that population migration is more motivated by economic interests, while people mobility has more individual choice differences.

Secondly, we have updated some of the older references, mainly in the review of the literature in the section on impact factors. The literature is too old to explain the impact factors of migration and intercity human flow in the context of rapid urban growth and renewal, so we have updated it to use recent literature to support our point.

3. Methods are advanced, appropriate, and accurate. However, add some detail to justify the application of these techniques in this study.

Response: Thanks for this suggestion. In response to the expert suggestion, we have added a section in subsection 3.4.4 about the MGWR model being the most feasible and effective in this study. First, we provide an introduction to the application of the GWR model, and we will and add some references to support the argument. Secondly, we provide a more detailed description of the MGWR model, which, although it is an innovative use of a technical approach, has been effectively applied in various fields. Therefore we are building on the previous expert study and applying the technique to a more detailed spatial unit in the context of the current study. It is an extension and refinement of previous research using the technique.

4. Be consistent with table headings. Recheck the table numbers.

Response: Thanks for this suggestion. In response to expert advice, we have checked the formatting and various sections of the entire paper to ensure that such problems do not occur again.

5. Add more citations in the explanations of results in discussion, also discuss the results more appropriately.

Response: Thanks for this suggestion. We have revised the conclusion section of 5.1. Firstly, we have discussed the results of each analysis in more detail by point, including the following three main points: (1) a summary of the spatial distribution characteristics of long-stay international visitors; (2) a summary of the spatial heterogeneity characteristics of each impact factor; and (3) a summary of the spatial characteristics of the impact factor coefficients after cluster analysis. Secondly, in the analysis of the causes of the result, we add factual arguments and include more citations to support our explanations.

6. Revise the conclusion to describe your study's problem, methods, and innovative findings. Add limitations of the study.

Response: Thanks for this suggestion. 

We have revised the discussion section of 5.2. In response to expert suggestions, we have reordered the lines and added sections as appropriate to express our views and the significance of this study more clearly.

Firstly, we added the discussion of the research object.

Secondly, we organize and explain the innovation points of this paper's research. The three main points are as follows: (1) A clearer discussion of the theoretical perspective of the study, which is based on the Space of Flow theory to discuss the travel behavior of long stay international visitors, rather than population migration theory. (2) Emphasis on the use of the MGWR model for the first time to study the impact of city attributes on the mobility patterns formed by the travel of this group of people. (3) This paper uses more timely and granular data, which is not possible with statistical data.

And we found some interesting phenomena that validate that the push-pull theory model does not use our research subjects. We find that the driving mechanism of long stay international visitors is not the same as that of local residents and short-term tourism behavior, citing literature to support our view, while suggesting that behind it reflects the issue of regional differences in the development of the Yangtze River Delta.

Finally, we discuss that the shortcoming of this study is that it has not been able to take into account the subjective willingness of individuals to travel, and that more in-depth research will be conducted from this perspective in the future.

7. Revise for typo mistakes.

Response: Thanks for this suggestion. We have further checked the text of the paper to ensure that such issues do not arise again.

Reviewer #2:

It is my pleasure to review the manuscript for the esteemed journal. After reading this manuscript, i am suggestion some minor changes that may further improve this manuscript.

Abstract: Abstract section is relatively lengthy,it should be short and to the point.

Response: Thanks for your positive comments on the study of this paper and for pointing out the existing shortcomings. We have taken your suggestions into account in this section and have made the relevant changes. The abstract section is not in line with the concise and easy-to-understand style of the journal, so we have reduced it to a short description of the aims of the study, the methods and content of the study and the results, and finally a brief statement of the significance of the study. The abstract section will be limited to around 200 characters.

Introduction, Literature review and methodology section is fine and very well written.

Response: Thank you for acknowledging the content of our research!

Result, discussion and conclusion: The arrangement of discussion and conclusion is not as per requirement of journal. Author first conclude the paper than include the discussion section, and this discussion section is also missing the support of previous studies.

Response: Thanks for this suggestion. In response to your suggestions, we have revised the conclusions in 5.1 and the discussion section in 5.2.

5.1 Conclusion

We have revised the conclusion section of 5.1. Firstly, we have discussed the results of each analysis in more detail by point, including the following three main points: (1) a summary of the spatial distribution characteristics of long-stay international visitors; (2) a summary of the spatial heterogeneity characteristics of each impact factor; and (3) a summary of the spatial characteristics of the impact factor coefficients after cluster analysis. Secondly, in the analysis of the causes of the result, we add factual arguments and include more citations to support our explanations.

5.2 Discussion

We have reordered the lines and added sections as appropriate to express our views and the significance of this study more clearly.

Firstly, we added the discussion of the research object.

Secondly, we organize and explain the innovation points of this paper's research. The three main points are as follows: (1) A clearer discussion of the theoretical perspective of the study, which is based on the Space of Flow theory to discuss the travel behavior of long stay international visitors, rather than population migration theory. (2) Emphasis on the use of the MGWR model for the first time to study the impact of city attributes on the mobility patterns formed by the travel of this group of people. (3) This paper uses more timely and granular data, which is not possible with statistical data.

And we found some interesting phenomena that validate that the push-pull theory model does not use our research subjects. We find that the driving mechanism of long stay international visitors is not the same as that of local residents and short-term tourism behavior, citing literature to support our view, while suggesting that behind it reflects the issue of regional differences in the development of the Yangtze River Delta.

Finally, we discuss that the shortcoming of this study is that it has not been able to take into account the subjective willingness of individuals to travel, and that more in-depth research will be conducted from this perspective in the future.

---

## [Editor Report · Decision Letter 1]

16 Aug 2023

Spatial Impacts of the Intercity Flow of Long Stay International Visitors Based on Mobile Phone Data in Yangtze River Delta, China

PONE-D-23-12368R1

Dear Dr. MENG,

We’re pleased to inform you that your manuscript has been judged scientifically suitable for publication and will be formally accepted for publication once it meets all outstanding technical requirements.

Kind regards,

Najabat Ali, Ph.D.

Academic Editor

PLOS ONE
---

## [Editor Report · Acceptance letter]

29 Aug 2023

PONE-D-23-12368R1 

Spatial Impacts of the Intercity Flow of Long Stay International Visitors Based on Mobile Phone Data in Yangtze River Delta, China 

Dear Dr. Meng:

I'm pleased to inform you that your manuscript has been deemed suitable for publication in PLOS ONE. Congratulations! Your manuscript is now with our production department. 

Kind regards, 

on behalf of

Dr. Najabat Ali 

Academic Editor

PLOS ONE